# Establishment of Novel Gastric Cancer Patient-Derived Xenografts and Cell Lines: Pathological Comparison between Primary Tumor, Patient-Derived, and Cell-Line Derived Xenografts

**DOI:** 10.3390/cells8060585

**Published:** 2019-06-14

**Authors:** Takeshi Kuwata, Kazuyoshi Yanagihara, Yuki Iino, Teruo Komatsu, Atsushi Ochiai, Shigeki Sekine, Hirokazu Taniguchi, Hitoshi Katai, Takahiro Kinoshita, Atsushi Ohtsu

**Affiliations:** 1Department of Pathology and Clinical Laboratories, National Cancer Center Hospital East, Kashiwa 277-8577, Japan; 2Division of Innovative Pathology and Laboratory Medicine, EPOC, National Cancer Center, Kashiwa 277-8577, Japan; 3Division of Biomarker Discovery, EPOC, National Cancer Center, Kashiwa 277-8577, Japan; kyanagih@east.ncc.go.jp (K.Y.); yuiino@east.ncc.go.jp (Y.I.); tkomatsu@east.ncc.go.jp (T.K.); aochiai@east.ncc.go.jp (A.O.); 4Department of Pathology and Clinical Laboratories, National Cancer Center Hospital, Tokyo 104-0045, Japan; ssekine@ncc.go.jp (S.S.); hitanigu@ncc.go.jp (H.T.); 5Department of Gastric Surgery, National Cancer Center Hospital, Tokyo 104-0045, Japan; hkatai@ncc.go.jp; 6Department of Gastric Surgery, National Cancer Center Hospital East, Kashiwa 277-8577 Japan; takkinos@east.ncc.go.jp; 7Director, National Cancer Center Hospital East, Kashiwa 277-8577, Japan; aohtsu@east.ncc.go.jp

**Keywords:** patient-derived xenograft, cell line, gastric cancer, pathology

## Abstract

Patient-derived xenograft (PDX) models have been recognized as being more suitable for predicting therapeutic efficacy than cell-culture models. However, there are several limitations in applying PDX models in preclinical studies, including their availability—especially for cancers such as gastric cancer—that are not frequently encountered in Western countries. In addition, the differences in morphology between primary, PDX, and tumor cell line-derived xenograft (CDX) models have not been well established. In this study, we aimed to establish a series of gastric cancer PDXs and cell-lines from a relatively large number of gastric cancer patients. We also investigated the clinicopathological factors associated with the establishment of PDX and CDX models, and compared the histology between the primary tumor, PDX, and CDX that originated from the same patient. We engrafted 232 gastric cancer tissues into immune-deficient mice subcutaneously and successfully established 35 gastric cancer PDX models (15.1% success rate). Differentiated type adenocarcinomas (DAs, 19.4%) were more effectively established than poorly differentiated type adenocarcinomas (PDAs, 10.8%). For establishing CDXs, the success rate was less influenced by histological differentiation grade (DA vs. PDA, 12.1% vs. 9.8%). In addition, concordance of histological differentiation grade between primary tumors and PDXs was significant (*p* < 0.01), while concordance between primary tumors and CDXs was not. Among clinicopathological factors investigated, pathological nodal metastasis status (pN) was significantly associated with the success rate of PDX establishment. Although establishing cell lines from ascites fluid was more efficient (41.2%, 7/17) than resected tissues, it should be noted that all CDXs from ascites fluid had the PDA phenotype. In conclusion, we established 35 PDX and 32 CDX models from 249 gastric cancer patients; among them, 21 PDX/CDX models were established from the same patients. Our findings may provide helpful insights for establishing PDX and CDX models not only from gastric but from other cancer types, as well as select preclinical models for developing new therapeutics.

## 1. Introduction

Cancer is the second leading cause of death worldwide and is estimated to have caused 9.6 million deaths in 2018. Among the various types of cancer, gastric cancer is the sixth most common cancer type and the third leading cause of cancer death. As with other types of cancer, advanced stage patients are treated with chemotherapy. However, chemotherapy has limited efficacy, and new therapeutic agents are urgently needed.

During the early preclinical stage of developing new therapeutics, it is necessary to employ appropriate in vitro or in vivo preclinical models. In this sense, cell culture models have been widely used because of their availability and suitability for mass handling. However, there are often discrepancies between results obtained with cell line models and those obtained from clinical trials. Therefore, cell line models are not considered the most appropriate model for predicting the efficacy of new drugs, presumably because of a lack of heterogeneity and tissue structure.

Patient-derived xenograft (PDX) models involve the implantation of tissue or cells from a patient’s tumor into an immunodeficient mouse [1,2]. The advantage of using PDXs is that they retain the tumor tissue architecture, which is absent in cell culture models [3]. Despite their limited availability and the difficulty in handling them, PDXs are considered a suitable preclinical model for evaluating drug sensitivity when developing new therapeutics, especially in oncology.

However, it is not clear if PDX models resemble the primary tumors from which they originated or how they do compare with CDX models. Therefore, it may be beneficial to directly compared PDX and cell line models established from the same tumor lesion. Another concern of using PDX models is their availability. For example, although there have been several studies using gastric cancer PDX models, most of them examined a relatively small number (less than 10) of PDX tumors [4,5,6,7,8,9,10]. Therefore, it is necessary to provide a series of PDX models for each type of cancer. It is also important to have PDX models validated, especially if histology of the xenograft is comparable with the parental tumor [11].

In this article, we describe 35 newly established gastric cancer PDX models as well as 32 gastric cancer CDX models. Among them, both PDX and CDX models were established from the same patient in 21 cases, allowing a direct histological comparison between primary, PDX, and CDX tumors. We also present clinicopathological factors associated with PDX and/or CDX establishment.

## 2. Materials and Methods

### 2.1. Patients

Two hundred fifty (250) gastric cancer patients treated at the National Cancer Center Hospital East (NCCHE) or the National Cancer Center Hospital (NCCH) from May 2013 to Feb 2017 were enrolled in this study. Two hundred thirty-two patients underwent surgical resection, and one (No. 214) was an autopsy case. One case (No. 238) harbored a malignant lymphoma in addition to a gastric cancer lesion, and malignant lymphoma tissue was taken for xenograft production. Therefore, the case was excluded from further analysis, and the remaining 232 cases including the autopsy case were designated as surgical cases. The remaining 17 cases received cell-free and concentrated ascites reinfusion therapy (CART) [12,13] for retaining ascites fluid caused by peritoneal dissemination of gastric cancer during the period and were designated as CART cases. CART is a method used for the treatment of ascites in patients who cannot receive higher doses of diuretics because of resistance to diuretic therapy or adverse effects (especially in patients with malignancies). Briefly, CART procedures were performed as follows: (i) drainage of the ascites by abdominal paracentesis; (ii) removal of malignant cells and other cell types by filtration and removal of excess fluid and electrolytes by concentration; and (iii) reinfusion of the filtered and concentrated ascites [12]. All 17 patients enrolled in the study had malignant ascites caused by peritoneal dissemination of gastric cancer cells, and approximately 3–5 L of ascites were collected from each patient. Patient characteristics of all study participants, except for the malignant lymphoma case, and of all surgical cases are summarized in Table 1 and Table 2, respectively. Clinicopathological data were obtained from hospital medical records. Written informed consent was obtained from all enrolled patients, and this study was approved by our institute’s Institutional Review Board (No. 2012-328).

### 2.2. Animals

Female NOD/SCID/gamma-c null (NOG) mice [14] were purchased from the Central Institute for Experimental Animals (CIEA, Kawasaki, Japan). Female ShiJic-scid Jcl (SCID) mice were purchased from CLEA Japan, Inc. (Tokyo, Japan). Female NOD.Cg-PrkdcscidIl2rgtm1Wjl/SzJ (NSG) mice were obtained from Charles River Laboratories International, Inc. (Kanagawa, Japan). All the mice were maintained in our animal facility under specific pathogen-free conditions. The experiments were performed according to the “Guidelines for Animal Experiments of the National Cancer Center” and were approved by the Institutional Ethics Review Committee for Animal Experimentation of the National Cancer Center (E21-M2-15, K16-004-M2).

### 2.3. Reagents

Dulbecco’s modified Eagle medium (DMEM), F-12 nutrient mixture (Ham’s F-12) and penicillin-streptomycin solution (PS) were purchased from Invitrogen (Carlsbad, CA, USA). Antibiotic-antimycotic mixed stock solutions (AAMS, 100×) were purchased from Nacalai Tesque Inc., (Kyoto, Japan). Fetal calf serum (FCS) was purchased from Biowest (Riverside, MO, USA). Cryopreservation reagents, Cell Banker, were purchased from Nippon Zenyaku Kogyo (Fukushima, Japan). Matrigel basement membrane matrix was purchased from Corning (Corning, NY, USA).

PAXgene tissue fix containers were purchased from QIAGEN (Tokyo, Japan). PATHWAY anti-HER-2/neu (4B5), INFORM HER2 Dual ISH DNA Probe Cocktail Assay (780-4422), and EBER 1 DNP Probe (760-1209) were purchased from Roche Diagnostics (Tokyo, Japan)

### 2.4. Establishing Xenograft Models

After examination by pathologists (T.K, S.S. or H.T.), about 25 mm^3^ of tumor tissue was resected from each primary lesion and transferred to tissue storage solution (DMEM supplemented with 10% FCS, 5% PS, and 5% AAMS). The tissue was washed extensively with wash solution (DMEM supplemented with 5% PS and 10% AAMS), dissected into 1-mm squares, and subcutaneously engrafted in NOG mice by using a transplantation needle (KN-391-20, Natume-seisakusho, Tokyo, Japan). NSG and SCID mice were used instead of NOD mice for 10 and 45 cases, respectively. When tumor tissues were engrafted into mice, matrigel was included in 63 cases (53 in NOG mice and all 10 cases in NSG mice).

The mice were observed daily for clinical signs and mortality. When subcutaneous tumors reached the size of 10 mm in diameter, the mice were sacrificed under anesthesia and the subcutaneous tumors were removed. The tumors were then dissected into pieces and either continuously engrafted to SCID mice, prepared for frozen stock, or fixed in PAXgene FIX for examining morphology.

For preparing CDXs, 10 million cells were subcutaneously engrafted into SCID mice, and subcutaneous tumors that arose were morphologically examined as described above.

### 2.5. Cell Culture

For patients who received CART, the cells originated in their ascites fluid and were trapped in the filtration filter during CART were recovered by washing in physiological saline. After centrifugation (2000 rpm, 10 min) and red blood cell removal using RBC lysis buffer (8.3% NH_4_Cl, 170 mM Tris-HCl (pH 7.65), cells were suspended in culture medium (50% DMEM, 50% Ham’s F-12) supplemented with 1% PS and 15% FCS. Tumor cells were seeded into 100 mm culture dishes (Falcon; Thermo Fisher Scientific, Waltham, MA, USA) and maintained at 37 °C in a humidified incubator with 5% CO_2_.

For establishing the cell lines from primary or xenograft tumors, tumor tissue was dissected into 1 mm cubic squares and explanted into 60 mm Corning Primaria dishes (Corning, NY, USA) with 1 mL of culture medium. An additional 4 mL culture medium was added the next day, and half the amount of culture medium was replaced twice weekly. Dishes containing tissue fragments were observed daily under an inverted phase microscope. The following three methods were used to selectively remove overgrowth fibroblasts: (i) trypsin treatment (0.05% trypsin and 0.02% EDTA, Thermo Fisher Scientifics): fibroblasts exfoliation by differences in trypsin sensitivity, with fresh medium added and washing performed to remove fibroblasts; (ii) physical treatment: change to serum-free medium and detachment of fibroblasts only using sharp silicone rubber under a microscope; and (iii) after exfoliating the cells using enzymatic treatment (Tumor Dissociation Kit # 130-095-929, Miltenyi Biotec, Tokyo, Japan), mouse-derived cells were removed using an antibody column (Mouse Cell Depletion Kit # 130-104-694, Miltenyi Biotec) according to the manufacturer’s protocol. When epithelial cell colonies were observed, they were transferred to another dish and continuously cultured. In some cases, we used the Mouse Cell Depletion Kit (Miltenyi Biotec, Tokyo, Japan) to remove mouse-derived cells. After 10 passages, the cells were considered an established line and subjected to short tandem repeat (STR)-analysis (Promega Japan, Tokyo, Japan). Cell lines were routinely tested for mycoplasma using a PCR mycoplasma detection technique at the Central Institute for Experimental Animals (Tokyo, Japan), and no contamination was detected.

### 2.6. Histology, Immunohistochemistry (IHC) and In Situ Hybridization (ISH)

Four micro-millimeter sliced tissue sections were prepared and subjected to H.E. staining, IHC, and ISH. Histological classifications were made based on the Japanese classification of gastric carcinoma (3rd English edition) [15], and differentiation grade was based on the presence/absence of glandular structure formation. IHC for HER2 expression and dual-IHC (DISH) assay for HER2 gene amplification were performed, as reported previously [16]. Criteria for deciding if cells were HER2-positive were either strong membranous staining of the HER2 protein or moderate membranous staining of HER2 plus HER2 gene amplification. ISH for Epstein-Barr virus (EBV)-encoded small RNA (EBER) was performed, as reported previously [17].

### 2.7. Statistics

Fisher’s exact test was performed using IBM SPSS Statistics version 22 (SPSS Inc., Chicago, IL, USA). The analyses performed were two-sided, and *p* < 0.05 was considered to be statistically significant.

## 3. Results

### 3.1. Establishment and Characterization of PDXs, Cell Lines, and CDX

From 232 surgical cases of gastric cancer, cancer tissues were subcutaneously engrafted into either NOG, NSG, or SCID mice. One hundred and fourteen subcutaneous tumors developed. However, after careful pathological examination, 74 tumors were confirmed as not of human epithelial cell origin and therefore excluded from further procedures (see Section 3.4). The remaining 40 subcutaneous tumors were considered primary xenograft tumors and continuously engrafted into SCID mice. Among 40 primary xenografts, 35 tumors were successfully transplanted into SCID mice through five passages and considered as established PDX models. The average number of weeks of tumor growth for xenograft expansion until tumor harvesting is shown in Appendix A. Among the remaining five primary xenograft tumors, one (No. 005) failed to develop into a tumor in re-engrafted SCID mice, three (No. 035, 096, and 249) developed into lympho-proliferative lesions in re-engrafted SCID mice, and one (No. 228) showed a slower growth rate and has not reached 5th passage as of the preparation of this manuscript (Figure 1).

During the process of establishing xenograft models, primary tumors (No. 005 and 096) or sequentially engrafted xenograft tumor tissues excised from SCID mice were used for establishing cell lines, and 23 cell lines were successfully established. In addition, another two cell lines were established from other cases, by directly culturing the primary cancer tissues, which failed to develop primary xenograft tumors, (Figure 1).

From 17 CART cases, four cell lines were established by directly culturing cells from ascites fluid. Another three cell lines were established by subcutaneously engrafting the primary cultured cells, and the subsequent subcutaneous tumor tissues were used as the source for further culturing (Figure 2).

In summary, we established 35 gastric cancer PDX models as well as 32 gastric cancer cell lines, 21 of which were established from the same patients.

### 3.2. Comparison of Histology between Primary, PDX, and CDX Tumors

First, we compared the histology between primary and PDX tumors. Thirty-five PDX models established were all adenocarcinomas, 28 of which had the same histological differentiation grade between primary and PDX tumors (Figure 3). Concordance between primary and PDX tumors was statistically significant (Table 3, *p* < 0.01). Although the remaining seven cases were classified as different histological differentiation grades, five cases showed mixed differential grades in either primary or PDX tumor lesions. Therefore, most cases (94.3%, 33/35) shared the same histology, at least in some regions, in both primary and PDX tumors. It should also be noted that tissue samples for PDX were primarily obtained from the mucosal/superficial areas of tumor lesions. In adenocarcinoma with mixed histology, differentiated histology is most frequently observed in the superficial area. Since HER2 expressions are routinely examined in gastric cancer as the target of Trastuzumab, we investigated HER2 expression in PDX and CDX from the case No.34 which showed HER2 positive in primary tumor. Strong Her2 expressions were observed in PDX and CDX (Figure 3).

However, concordance between primary and CDX tumors showed certain inconsistencies. Among 24 cases in which CDXs were obtained from resected tissue, only about half the cases were of the same differentiation grade (Table 4). It is noteworthy that more than half the cases in which primary tumors showed differentiated histology (differentiated adenocarcinomas, DAs) became poorly differentiated adenocarcinomas (PDAs) in CDXs.

Among the seven cell lines established from the CART cases, six of them developed CDX tumors in SCID mice. Interestingly, all CDXs from CART cases were of the poorly differentiated phenotype (Figure 4).

### 3.3. Consistent Histology after Sequential Transplantation of PDXs

In three cases (No.14, 17, 34), all of which were of the differentiated phenotype in both primary and PDX tumor lesions at 5th passage, PDX tumor tissues were continuously re-engrafted into other SCID mice 15 times. Impressively, all cases conserved the differentiated phenotype at the time of the 15th passage. In case No.14, although both primary and PDX tumor lesions were of the differentiated phenotype, the CDX tumor lost its tubular structure and was classified as having a poorly differentiated phenotype (Figure 5).

### 3.4. Development of Lymphoproliferative Lesions and Tumors of Mouse Cell Origin

As described above, in this study, we observed lymphoproliferative lesions (LPLs) at the site of gastric cancer tissue engraftment with relatively high frequencies. As lymphoma transformation has been reported to occur in NOG mice [7], we evaluated NSG and SCID mice but found that LPLs developed at similar frequencies in all three models: NOG 31.0% (55/177), NSG 30.0% (3/10), and SCID 22.2% (10/45). Although the occurrence of LPLs was slightly lower in SCID than in NOG mice, the difference was not statistically significant. In 114 subcutaneous tumors that developed, more than half of the tumors were LPLs. To determine if LPLs were of human lymphocyte origin, we examined whether human chromosome 17 (hCr 17) was present using the DISH assay. All LPLs were positive for the hCr17, thus we concluded that the proliferating lymphocytes are of human origin. For most cases, the majority of lymphocytes were CD20-positive B cells, with lesser numbers of CD3-positive T cells (Figure 6). Since EBV is known to transform human B cells, we examined whether the proliferating B cell harbored EBV. All cases, except for one, were EBER-negative. Thus, B cells proliferating in the mouse subcutaneous tissue were not EBV-transformed. The frequency of developing LPLs was not affected by the mouse phenotype, and LPLs were observed even in SCID mice (Table 5).

In addition, five cases developed malignant tumors with mouse epithelial-cell origin, confirmed by the absence of hChr 17. All tumors developed in NOG mice, and matrigel was inoculated with primary gastric cancer tissues in all except one case.

### 3.5. Pathological Factors Affected in the Establishment PDX/Cell Line

Clinicopathological factors associated with the establishment of PDXs were investigated. In 322 surgical cases, 34 cases received chemotherapy prior to surgery. The establishment rate for PDXs was higher in those cases that received chemotherapy than in those that did not (26.4% (9/34) vs. 13.1% (26/198), respectively), although the difference was not statistically significant. Among 232 cases in the surgical group, 226 cases showed pure adenocarcinoma histology (Table 6). In this subgroup, the PDX establishment rate was higher for differentiated subtypes compared with poorly differentiated subtypes (19.4% (24/124) vs. 10.8% (11/102)), respectively. It is also noteworthy that for the other six cases that included mixed adenoneuroendocrine and squamous cell carcinomas, xenograft tumors failed to develop in mice.

For the 198 surgical cases without prior chemotherapy, pathological stage, TNM factors as well as histological differentiation grade were evaluated for their association with the PDX establishment rate (Table 7). Interestingly, the pN factor, which is indicative of lymph node metastatic status, was associated with the PDX establishment rate, while other pathological factors did not show significant association. To mitigate for the possibility that the results were influenced by the type of immunodeficient mouse model used to establish PDX, we stratified the data by the type of mouse model and repeated the analysis. The pN factor, but not pT nor pM, was still associated with the establishment rate of PDX (Appendix A). Although the reason why pM did not show a significant correlation with the PDX establishment rate remains to be clarified; one possibility is that pM is associated with PDA phenotype as the main cause of pM in gastric cancer is peritoneal dissemination, which is frequently observed in the PDA phenotype. Thus, we also reanalyzed the data after stratification by histological differentiation grade and found that pM again failed to show an association with PDX success rate, even in the DA subgroup (Appendix A). Again, tumors with a differentiated phenotype resulted in a higher PDX establishment rate than poorly differentiated ones in this subgroup, but the difference was not statistically significant. To investigate the possibility whether the location of tumor originated from is associated with successful PDX establishment, we analyzed the data. Interestingly, establishment success rate is highest in esophagogastric junction cases (20.0%) followed by lower (15.1%), middle (10.2%), and upper portion (8.9%) of the stomach (Appendix A). The mechanism which causes these differences have yet to have been clarified. There were no factors identified that significantly affected the CDX establishment rate.

## 4. Discussion

PDX models have recently been widely accepted as a preclinical model [1,2,18,19,20,21,22,23]. However, the utility of PDX models has not yet been fully exploited in most research laboratories, mainly because of the difficulty of accessing and maintaining PDXs. It is especially apparent for cancer types, including gastric cancer, not frequently encountered in Western countries [11]. Although there have been several studies reporting gastric cancer PDX models [4,5,6,7,8,9,10,24,25,26,27], a relatively small number of PDX models were examined in most of the cases. Therefore, it is urgently necessary to establish relatively large series of gastric cancer PDX models to obtain more accurate results from preclinical studies for predicting the efficacy of newly developed drugs specific to gastric cancer. The 35 gastric cancer PDX models we established in this study may contribute to solving this issue (summarized in Table 8).

It has been considered that PDX models are biologically more similar to primary tumors than to CDX models. However, to our best knowledge, there have been no reports available describing the difference between primary, PDX, and CDX tumors obtained from the same primary gastric cancer lesion. In fact, one report argued that there was discordance between the histology of primary tumors and PDXs [24]. In the present study, we have shown that most PDX tumors showed histologically consistent morphology with the primary tumors. Conversely, more than half of the CDX tumors showed discordance with the primary tumors in terms of histological differentiation grade. Although there may be several reasons to account for this discordance, one possibility is that the most aggressive and dedifferentiated clone(s) with higher proliferation activity, which are associated with poorly differentiated adenocarcinoma characteristics, take over the other clones during the culturing process for establishing cell lines. We also note the possibility that the relatively higher concentration of FCS (15%) used in the present study may have promoted the proliferation of more aggressive clones, although further studies are needed to confirm this. It is also noteworthy that PDX models were established more readily from differentiated adenocarcinomas, while cell line/CDXs were established more readily from poorly differentiated adenocarcinomas. Although the reason why differentiated adenocarcinoma was more readily established than poorly differentiated adenocarcinoma remains to be elucidated, one possibility is that the former places fewer demands on stromal cells. For example, poorly differentiated adenocarcinoma is known to utilize TGF-beta during stromal reactions, although human TGF-beta may not interact with mouse stromal cells.

It was reported previously that the success rate of PDX establishment is higher with more advanced lesions [28]. In this study, we showed that lesions having LN metastases had a higher success rate compared with lesions without metastases. To our surprise, the existence of distant metastases (pM) failed to show a significant association with success rate. For establishing gastric cancer cell lines, the cells recovered from ascites fluid in CART cases showed a higher success rate than surgical cases. However, it should be noted that all of the CDXs derived from CART cases had a poorly differentiated phenotype.

In this study, a strong negative factor interfering with the establishment of PDXs was the development of LPLs. Previously, it was reported that 33.3% of these gastric cancer tissues developed lymphomas in NOG mice [7]. In this present study, LPLs were observed at similar frequencies (30.6%, 61/272). Although the previous study reported that lymphoma development was specific to NOG mice, in this study, LPLs developed in not only NOG but also in SCID mice. More recently, Corso et al. reported the occurrence of lymphoma after gastric cancer tissue engraftment in NOD/SCID mice [29]. We further identified that most of the LPLs were derived from B lymphocytes and were not EBV-associated, although Corso et al. observed that the lymphomas that developed in their study were EBV-associated. We do not deny the potential contribution of EBV in LPLs in the present study, however, as at least one of the cases showed diffuse EBER expression in proliferating B cells. Further studies will be needed to determine what percent of LPLs observed in this study are a clonal disease and whether the developing LPLs are specific for gastric cancer.

Our study has some limitations. First, intratumoral heterogeneity was not evaluated in this study. Second, because molecular classification is applied in various types of cancer, including gastric cancer [30], it would be interesting to examine whether the molecular profiles were conserved between primary, PDX, and CDX tumors [25]. Here, we have presented only a single case in which HER2 overexpression was observed in the primary, PDX, and CDX tumors. Other molecular profiles are under investigation and will be presented in the future. Third, none of the PDX or CDX models developed metastatic lesions in mice. Therefore, the models are not suitable for examining the mechanism of tumor metastasis. Fourth, since immunodeficient mice are required for the development of PDX/CDX models, these models are not suitable for the evaluation of immunotherapies unless a humanized immune system is restored by co-transplantation of immune cells from the same patient.

## Figures and Tables

**Figure 1 cells-08-00585-f001:**
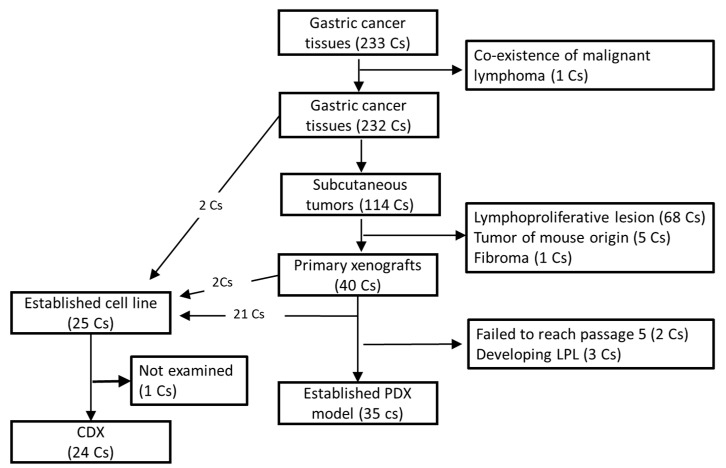
Diagram of PDXs and Cell lines established from surgical cases. Out of 232 gastric cancer tissues, 35 PDX models as well as 25 gastric cancer cell lines were established. Twenty-four cell lines formed CDX tumors in SCID mice. Cs: Cases.

**Figure 2 cells-08-00585-f002:**
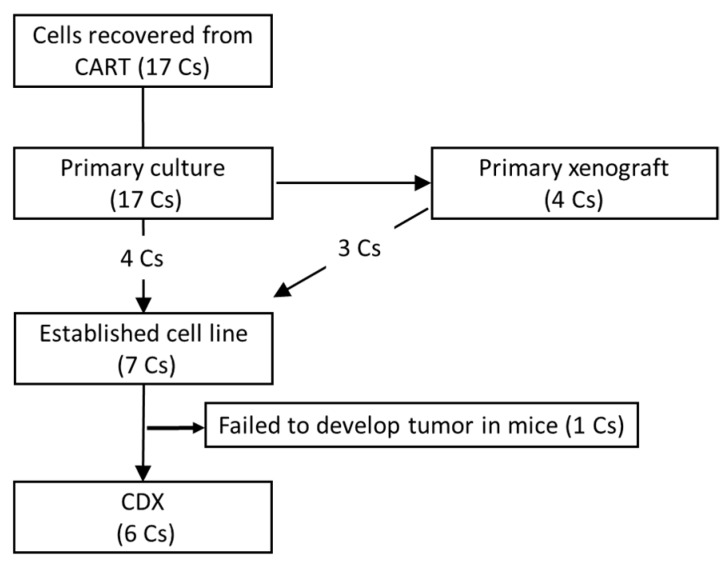
Diagram of cell lines established from CART cases. Out of 17 patients, seven gastric cancer cell lines were established. Six gastric cancer cell lines from CDX tumors in SCID mice. Cs: Cases.

**Figure 3 cells-08-00585-f003:**
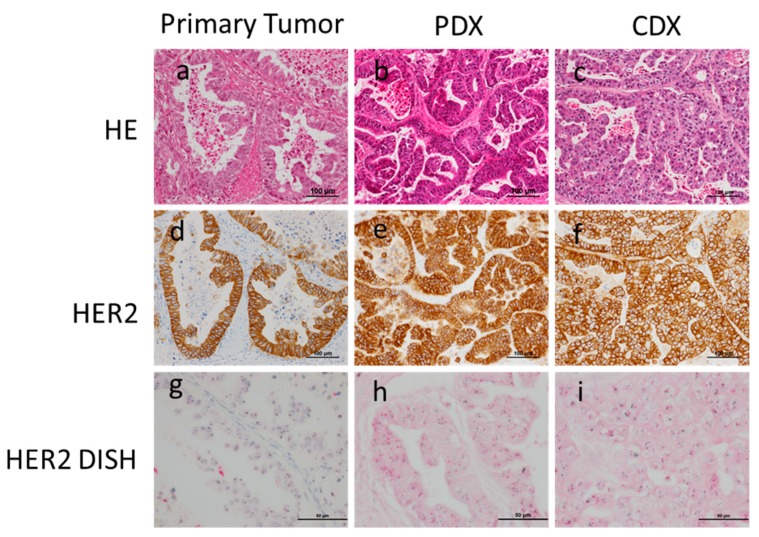
Histology of primary, PDX, and CDX tumors from a surgically resected specimen. (**a**–**c**) Primary, PDX, and CDX tumors showed similar papillary structures. (**g**–**i**) All tumors showed HER2 overexpression (**d**–**f**) as well as HER2 gene amplification.

**Figure 4 cells-08-00585-f004:**
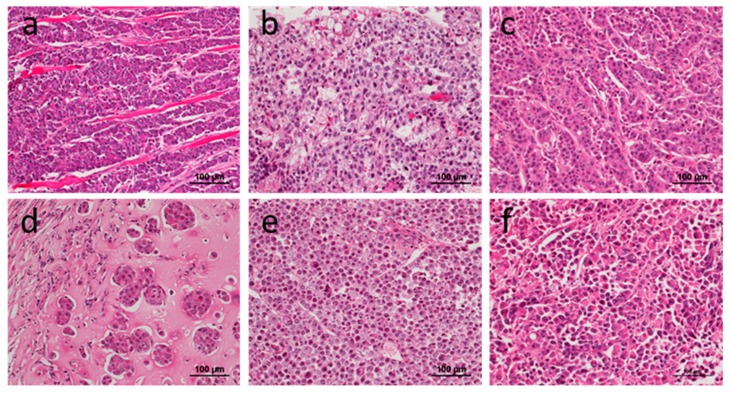
Histology of six CDXs from CART cases (**a**–**f**). All tumors were of the poorly differentiated adenocarcinoma phenotype.

**Figure 5 cells-08-00585-f005:**
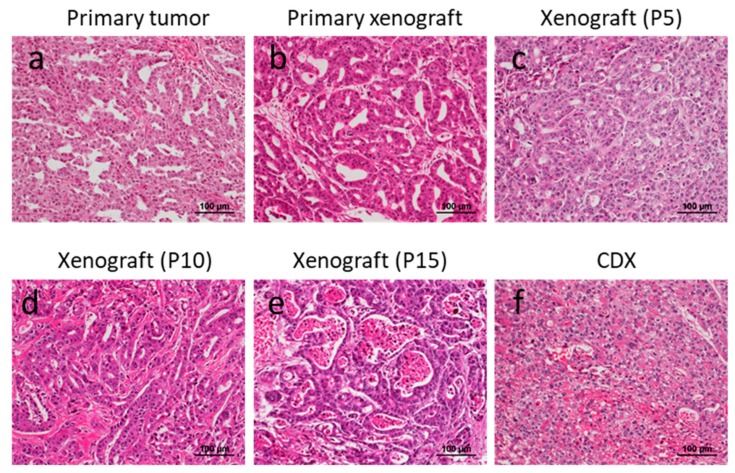
Comparison of histology after sequential re-engraftment into SCID mice. A primary xenograft tumor (**b**) as well as sequentially engrafted xenograft tumors (**c**–**f**) morphologically resemble the primary tumor (**a**).

**Figure 6 cells-08-00585-f006:**
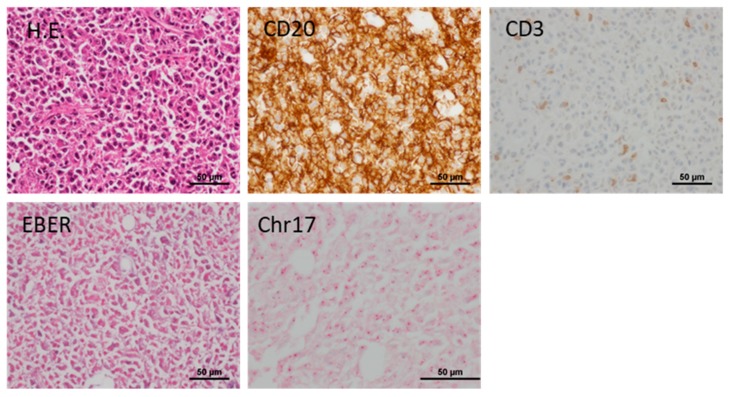
Histology and immuno-molecular phenotypes of lymphoproliferative lesions. The majority of mononuclear cells with moderate nuclear atypia were CD20-positive. They were positive for human Chromosome 17 probes but negative for EBER. A representative case is shown.

**Table 1 cells-08-00585-t001:** Patient Characteristics of all study participants (includes surgical cases).

	*n* = 249
Hospital (NCCHE/NCCH)	218/31
Age (Mean, Range)	67.2, 31–90
Sex (Male/Female)	160/72
Treatment (Surgery/CART)	232/17

NCCHE: National Cancer Center Hospital East, NCCH: National Cancer Center Hospital, CART: Cell-free and concentrated ascites reinfusion therapy.

**Table 2 cells-08-00585-t002:** Patient Characteristics of only surgical cases.

	*n* = 232
Hospital (NCCHE/NCCH)	201/31
Age (Mean, Range)	67.7, 31–90
Sex (Male/Female)	160/72
Specimen (Primary/Metastatic)	231/1
Histology of primary tumor (DA/PDA/Other) *	124/102/6
Prior chemotherapy (Not Received/ received)	198/34
Pathological stage **	
Patient who did not receive chemotherapy	
pStage (1/2/3/4)	40/63/90/5
pT (1/2/3/4)	30/26/69/73
pN (0/>1)	61/137
pM (0/1)	193/5
Histology of primary tumor (DA/PDA/Other) *	105/89/4
Patient who received chemotherapy	
ypStage (0/1/2/3/4)	2/3/10/10/9
ypT (0/1/2/3/4)	1/2/3/15/12/1
ypN (0/>1/ND)	6/22/6
ypM (0/1/ND)	1/7/27
Histology of primary tumor (DA/PDA/Other) *	19/13/2

* DA: Differentiated adenocarcinoma, PDA: Poorly differentiated adenocarcinoma, “Others” include mixed adenoneuroendocrine carcinoma, squamous cell carcinoma and unavailable cases. ** For “Pathological Stage”, pathological findings were presented separately as 2 subgroups; “Patient who did not receive chemotherapy” and “Patient who received chemotherapy”.

**Table 3 cells-08-00585-t003:** Concordance of histological differentiation grade between Primary and PDX tumors.

		PDX	
		DA	PDA
Primary Tumor	DA	22	2
	PDA	5	6
			*p* < 0.01

DA: Differentiated adenocarcinoma, PDA: Poorly differentiated adenocarcinoma.

**Table 4 cells-08-00585-t004:** Concordance of histological differential grade between Primary and CDX tumors.

		CDX	
		DA	PDA
Primary Tumor	DA	6	8
	PDA	2	8
			*p* = 0.39

DA: Differentiated adenocarcinoma, PDA: Poorly differentiated adenocarcinoma.

**Table 5 cells-08-00585-t005:** Outcomes of engrafted gastric cancer tissue in NOG, NSG, and SCID mice.

			Histology of Primary Xenograft Tumor (*n*)				
Mouse	Matrigel	Total Cases (*n*)	No Tumor	Carcinoma	LPL	Mouse Tumor	Fibroma
NOG	-	125	60	21	42	1	1
	+	52	25	10	13	4	0
NGS	+	10	5	2	3	0	0
SCID	-	45	28	7	10	0	0

LPL: Lymphoproliferative lesion.

**Table 6 cells-08-00585-t006:** Histological subtypes of Primary tumor and PDX.

Histology of Primary Tumor	Primary Tumor (*n*)	PDX (*n*)	PDX Establishing Rate (%)
DA	124	24	19.4
PDA	102	11	10.8
MANEC	3	0	0.0
SCC	2	0	0.0
ND	1	0	0.0

DA: Differentiated adenocarcinoma, PDA: Poorly differentiated adenocarcinoma, MANEC; Mixed neuroendocrine carcinoma, SCC: Squamous cell carcinoma, ND; Not determined.

**Table 7 cells-08-00585-t007:** Pathological factors related to PDX establishment.

	PDX		
	Failed	Established	*p*-Value
pStage			
I/II	86	17	0.21
III/VI	86	9	
pT			
1/2	47	9	0.49
3/4	125	17	
pN			
0	58	3	0.02
1<	114	23	
pM			
0	167	26	1.00
1	5	0	
Histology			
DA	87	18	0.14
PDA	81	8	

**Table 8 cells-08-00585-t008:** Summary of clinicopathological characteristics of primary, PDX, and CDX tumors.

	Primary Tumor				PDX		Cell Line	CDX
Case	Prior Chemotherapy	(y)pStage	Histology	HER2	Histology	HER2		Histology
013	NR	2	DA	N	DA	N	NE	-
014	R	4	DA	N	DA	N	E	PDA
017	NR	2	DA	N	DA	N	NE	-
033	NR	3	DA	N	DA	N	E	DA
034	NR	3	DA	P	DA	P	E	DA
038	NR	2	DA	N	DA	N	E	PDA
041	NR	3	DA	N	DA	N	E	DA
042	NR	3	DA	N	DA	N	NE	-
043	NR	3	DA	N	DA	N	E	PDA
056	NR	2	DA	N	DA	N	E	PDA
058	R	4	PDA	N	DA	N	NE	-
081	NR	2	DA	N	DA	N	E	PDA
086	NR	1	PDA	N	DA	N	E	PDA
093	R	2	DA	N	PDA	N	E	PDA
115	NR	3	PDA	N	PDA	N	NE	-
119	NR	2	DA	N	DA	N	NE	-
121	NR	1	DA	N	DA	P	NE	-
140	NR	3	PDA	N/A	DA	N	E	PDA
144	NR	1	DA	N	DA	N	NE	-
145	NR	3	PDA	N/A	PDA	N	NE	-
156	R	2	DA	N	DA	N	E	DA
159	R	3	PDA	N	DA	N	E	DA
165	NR	2	PDA	N	PDA	N	E	PDA
174	NR	3	PDA	N	PDA	N	E	PDA
175	NR	1	DA	N/A	DA	P	NE	-
193	R	4	DA	N/A	DA	N	NE	-
194	R	2	DA	N/A	PDA	N	NE	-
197	R	3	DA	N/A	DA	P	NE	-
202	NR	2	PDA	N	DA	N	E	DA
205	NR	2	DA	N/A	DA	N	E	DA
214	R	4	PDA	N/A	PDA	N	E	PDA
223	NR	1	DA	N	DA	N	E	PDA
234	NR	2	DA	N	DA	N	NE	-
241	NR	2	DA	P	DA	P	E	-*
245	NR	2	PDA	N	PDA	N	E	PDA

R; Received, NR; Not received, N; Negative, P; Positive, N/A; not available, E: Established, NE; Not established. *; CDX not examined.

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
