# Peer review of "Establishment of Novel Gastric Cancer Patient-Derived Xenografts and Cell Lines: Pathological Comparison between Primary Tumor, Patient-Derived, and Cell-Line Derived Xenografts"

_cells, 2019, doi:10.3390/cells8060585_

Round 1
Reviewer 1 Report
In this paper the authors establish 35 gastric cancer PDX models and 24 CDX models from 233 human gastric cancer tissues. 21 of the PDX and CDX models came from the same patient. The amount of work put into establishing these models is impressive and will be useful to the research community, but many justifications to the experimental design need to be added, details should be added to the CART cases for a better understanding, much of the immunohistochemistry (IHC) data is not displayed in a comprehensive manner, and the most important and interesting finding (that pN was the strongest correlator to engraftment) is not explored in detail.
To start with justifications, what is the purpose of creating both PDX and CDX models? Why not just create PDX models since the introduction claims these are most similar to the original tumors with respect to relevant factors such as drug response? Also, why are three different immunodeficient mice used to create PDX models? What is the benefit? Table 6 shows the differences in outcomes of engrafted tissue between the three mice but there is no statistical analysis of this, and no single patient-derived tumor seems to have been subject to engraftment with all three mice for a better comparison.
It is hard to follow what exactly happened for the 17 patients with cells recovered from CART. At which point were the cells isolated? Before, during or after CART therapy? Based off reading the methods, it appears the ascites fluid was recovered because it is expected to have circulating tumor cells. Next techniques were put in place to deplete the gathered cell population of fibroblasts and red blood cells and the remaining cell pool was assumed to be circulating gastric cancer cells mostly. Was there any confirmation to show that these cells including the cells shown in Figure 4 are mostly human gastric cancer in origin? Has a method like this ever been used before establish gastric cancer PDX or CDX models and if so can you reference it?
All of the IHC data shown is lacking scale bars and these should be added. The HER2 DISH in Figure 3 is not large enough nor at a high enough resolution to demonstrate the purpose of the assay. The reader should be able to look at these photos and count the copy numbers in a specific cell.
Finally, the most interesting and relevant data is that out of all the patient characteristics, pN correlated the most with successful engraftment, even more than pM. More tables need to be created exploring this difference. These additional figures should still include the pN status and whether or not the tumors engrafted, but also attempt to tease apart the potential confounding effects of mouse type used for engraftment and the impact of surgical resection verse the CART cell isolation techniques. For example it would be relevant to know if all of your CART derived cells fell into one category when looking at pN and engraftment. To make room in the paper, Figures 5 and 6 could be moved to a supplemental section.
Author Response
Response to Reviewer 1 Comments
Point 1: To start with justifications, what is the purpose of creating both PDX and CDX models? Why not just create PDX models since the introduction claims these are most similar to the original tumors with respect to relevant factors such as drug response?
Response 1: Thank you for your comment on the rationale for the present study. Along with exploring the factors associated with establishment success rate, we aimed to evaluate the histological differences between primary tumors and PDX/CDX models. Therefore, we sought to establish PDX models and cell lines simultaneously from the same patients. As the results of the study show, we successfully concluded that cell lines tended toward a more de-differentiated phenotype, indicating that the use of PDX models may be more advantageous than using CDX models. Our findings may be of relevant to researchers studying differentiated gastric adenocarcinomas.
To clarify this rationale, the Introduction section of the manuscript was revised as follows:
Line 62
“Therefore, it may be beneficial to directly compared PDX and cell line models established from the same tumor lesion.”
Line 71
“Among them, both PDX and CDX models were established from the same patient in 21 cases, allowing a direct histological comparison between primary, PDX, and CDX tumors.”
Point 2: Also, why are three different immunodeficient mice used to create PDX models? What is the benefit? Table 6 shows the differences in outcomes of engrafted tissue between the three mice but there is no statistical analysis of this, and no single patient-derived tumor seems to have been subject to engraftment with all three mice for a better comparison.
Response 2: The primary purpose for using two different immunodeficient mouse models (SCID and NSG) other than NOG mice was to determine whether the development of lympho-proliferative lesions (LPLs) could be prevented in these models. Although Choi et al. previously reported that lymphoma transformation occurred only in NOG mice and not in nude mice, we nevertheless encountered LPLs in both SCID and NSG mice. Although the frequency of LPLs was lower in SCID mice than in NOG mice, the difference was not significant. As the quantity of tumor tissues available for this study was limited, we were unable to simultaneously compare the outcomes for the three different immunodeficient mouse models. Nevertheless, we consider our findings to be relevant for researchers evaluating different immunodeficient mouse models to avoid the development of LPL. The text in section 3.4 of the Results has been modified accordingly, as follows:
Line 258
“As described above, in the present study, we observed lymphoproliferative lesions (LPLs) at the site of gastric cancer tissue engraftment with relatively high frequencies. As lymphoma transformation has been reported to occur in NOG mice, we evaluated NSG and SCID mouse models but found that LPLs developed at similar frequencies in all three models: NOG 31.0% (55/177), NSG 30.0% (3/10), and SCID 22.2% (10/45). Although the occurrence of LPLs was slightly lower in SCID than in NOG mice, the difference was not statistically significant.”
Point 3: It is hard to follow what exactly happened for the 17 patients with cells recovered from CART. At which point were the cells isolated? Before, during or after CART therapy? Based off reading the methods, it appears the ascites fluid was recovered because it is expected to have circulating tumor cells. Next techniques were put in place to deplete the gathered cell population of fibroblasts and red blood cells and the remaining cell pool was assumed to be circulating gastric cancer cells mostly. Was there any confirmation to show that these cells including the cells shown in Figure 4 are mostly human gastric cancer in origin? Has a method like this ever been used before establish gastric cancer PDX or CDX models and if so can you reference it?
Response 3: We apologize for not providing sufficient information about cell-free and concentrated ascites reinfusion therapy (CART) and the subsequent procedures. CART is a method used for the treatment of ascites in patients who cannot receive higher doses of diuretics because of resistance to diuretic therapy or adverse effects (especially in patients with malignancies). CART has been approved in Japan for use in clinical settings, and its safety and efficacy in patients with malignant ascites has been established [Hanafusa et al., PLoS ONE 12(5): e0177303].
The CART procedure used in the present study was as follows: i) drainage of the ascites by abdominal paracentesis; ii) removal of malignant cells and other cell types by filtration and removal of excess fluid and electrolytes by concentration; and iii) reinfusion of the filtered and concentrated ascites [Ref. 12]. All 17 patients enrolled in the study had malignant ascites caused by peritoneal dissemination of gastric cancer cells, and approximately 3–5 L of ascites were collected from each patient. After CART, the filtration filters were washed in physiological saline to recover the trapped cells for cultivation. As the ascites were the result of peritoneal dissemination of gastric cancer, recovered cells should include gastric cancer cells, although we did not examine the percentage of cancer cells among the total cell types.
We have updated the Materials and Methods section of the revised manuscript as shown below.
Line 82
“The remaining 17 cases underwent cell-free and concentrated ascites reinfusion therapy (CART) for retaining ascites fluid caused by peritoneal dissemination of gastric cancer during the study period and were designated as CART cases. CART is a method used for the treatment of ascites in patients who cannot receive higher doses of diuretics because of resistance to diuretic therapy or adverse effects (especially in patients with malignancies). Briefly, CART procedures were performed as follows: i) drainage of the ascites by abdominal paracentesis; ii) removal of malignant cells and other cell types by filtration and removal of excess fluid and electrolytes by concentration; and iii) reinfusion of the filtered and concentrated ascites. All 17 patients enrolled in the study had malignant ascites caused by peritoneal dissemination of gastric cancer cells, and approximately 3–5 L of ascites were collected from each patient.”
Line 137
“For patients who received CART, cells that originated in ascites fluid and were trapped in the filtration filter during CART were recovered by washing in physiological saline.”
Point 4: All of the IHC data shown is lacking scale bars and these should be added. The HER2 DISH in Figure 3 is not large enough nor at a high enough resolution to demonstrate the purpose of the assay. The reader should be able to look at these photos and count the copy numbers in a specific cell.
Response 4: In Figures 3, 4, 5, and 6, all photographic images of immunohistochemistry (IHC), H.E. staining, and dual-IHC (DISH) were re-captured and presented with a scale bar. Higher magnification was used for the new DISH images as well as for the image in Figure 6.
Point 5: Finally, the most interesting and relevant data is that out of all the patient characteristics, pN correlated the most with successful engraftment, even more than pM. More tables need to be created exploring this difference. These additional figures should still include the pN status and whether or not the tumors engrafted, but also attempt to tease apart the potential confounding effects of mouse type used for engraftment and the impact of surgical resection verse the CART cell isolation techniques. For example it would be relevant to know if all of your CART derived cells fell into one category when looking at pN and engraftment. To make room in the paper, Figures 5 and 6 could be moved to a supplemental section.
Response 5: Thank you for highlighting one of the most relevant findings in this study, that pN factor is associated with the success rate for PDX establishment. First, we wish to explain that the data contributing to this result are derived from the cohort of surgical cases without pre-operative chemotherapy. Therefore, no CART cases were included. Regarding the types of mouse models, we reanalyzed the data with stratification by mouse model. Again, pN was significantly associated with the PDX establishment rate, while pT and pM were not. In NSG and SCID mice, however, the difference was not statistically significant because the number of cases assigned to each of these models was relatively small.
The reason why pM did not show a significant correlation with PDX establishment rate remains to be clarified. As the most frequent cause of pM in gastric cancer is peritoneal dissemination, which is more often related to the poorly differentiated adenocarcinoma (PDA) phenotype than to the differentiated adenocarcinoma (DA), we assumed that a high frequency of PDA reduced the effect of pM on the PDX establishment rate. Thus, we also reanalyzed the data with stratification by histological differentiation grade. Again, pM failed to show an association with PDX success rate, even in the PA subgroup.
To explain these findings, we added the following sentences to the Results section, and also created Table S2 and Table S3.
Line 297
“To mitigate for the possibility that the results were influenced by the type of immunodeficient mouse model used to establish PDX, we stratified the data by the type of mouse model and repeated the analysis. The pN factor, but not pT nor pM, was still associated with the establishment rate of PDX (Table S2). Although the reason why pM did not show a significant correlation with the PDX establishment rate remains to be clarified, one possibility is that pM is associated with PDA phenotype as the main cause of pM in gastric cancer is peritoneal dissemination, which is frequently observed in the PDA phenotype. Thus, we also reanalyzed the data after stratification by histological differentiation grade and found that pM again failed to show an association with PDX success rate, even in the DA subgroup (Table S3).”
Table S2 Pathological factors related to PDX establishment (stratified by type of immunodeficiency mouse model)
Mouse | pT/N/M | PDX | p-value | ||
Fail | Success | ||||
NOG | pT | 1/2 | 38 | 7 | 0.62 |
3/4 | 95 | 14 | |||
pN | 0 | 47 | 2 | 0.02 | |
1< | 86 | 19 | |||
pM | 0 | 128 | 21 | 1.00 | |
1 | 5 | 0 | |||
NGS | pT | 1/2 | 0 | 0 | - |
3/4 | 5 | 0 | |||
pN | 0 | 1 | 0 | - | |
1< | 4 | 0 | |||
pM | 0 | 5 | 0 | - | |
1 | 0 | 0 | |||
SCID | pT | 1/2 | 9 | 2 | 0.61 |
3/4 | 25 | 3 | |||
pN | 0 | 10 | 1 | 1.00 | |
1< | 24 | 4 | |||
pM | 0 | 34 | 5 | - | |
1 | 0 | 0 | |||
Table S3 Pathological factors related to PDX establishment (adenocarcinoma cases only, stratified by type of histological differentiation grade)
Histology of primary tumor | pT/N/M | PDX | p-value | ||
Fail | Success | ||||
DA | pT | 0 | 30 | 6 | 0.10 |
1 | 57 | 12 | |||
pN | 0 | 34 | 2 | 0.03 | |
1 | 53 | 16 | |||
pM | 0 | 86 | 18 | 1.00 | |
1 | 1 | 0 | |||
PDA | pT | 1/2 | 15 | 3 | 0.20 |
1 | 66 | 5 | |||
pN | 0 | 23 | 1 | 0.44 | |
1 | 58 | 7 | |||
pM | 0 | 77 | 8 | 1.00 | |
1 | 4 | 0 | |||
DA: Differentiated adenocarcinoma, PDA: Poorly differentiated adenocarcinoma
Reviewer 2 Report
Overview: This manuscript describes the establishment of a rather large library of gastric cancer patient-derived models including xenografts (PDX) as well as cell-line derived xenografts (CDX). This represents one of the larger reported PDX libraries. This descriptive manuscript identifies clinicopathological features that associate with tumor establishment while describing some of the technical challenges that exist for PDX development from gastric cancer. While the reporting of this information is useful, the lack of molecular characterization mildly dampens the impact of the study. Some specific elements are commented on below:
Specific Comments:
1. There are some recent relevant papers that should be mentioned in the discussion. Most notably the Corso et al. Neoplasia 2018 (PMCID: 5915970) which describes the use of short term rituximab treatment at time of mouse implant which effectively prevents the development of lymphomas in gastric carcinoma PDX implants. Since the authors describe a similar phenomenon, the Corso paper is a critical reference for this manuscript. Other recent published papers that could be referenced include: Ryu et al. Cancers 2019 (PMID: 30965636) and the Liu and Meltzer review paper in Cellular and Molecular Gastroenterology and Hepatology 2017 (PMCID: 5404028).
2. The authors utilized high percentage fetal calf serum in the preservation media and in the cell culture conditions. It is expected that the FCS inclusion will promote proliferation due to the high growth factor content. The authors should acknowledge or discuss that this could be related to differences between the CDX and PDX development.
3. Another discussion point that is underdeveloped here is the paradoxical finding that the well differentiated adenocarcinomas were more readily established than poorly differentiated whereas the highly successful CDX were all poorly differentiated. Moreover, more local-regionally advanced tumors (node positive) were more likely to form PDX. These potentially counter-intuitive findings could be discussed better.
4. It would be good to know how much time occurred between last dose of chemotherapy and surgical resection for those patients who received chemotherapy. There are some PDX groups that recommend a minimum of 3 weeks between chemo administration and tumor cell harvesting for PDX implantation. The authors should provide this info if available.
5. Have the authors considered using a less immunocompromised mouse such as athymic nude mice for the host organism? Perhaps those mice would be less likely to form lymphomas.
6. Table 8 needs to be formatted better with words being wrapped into two lines for many of the columns.
Author Response
Response to Reviewer 2 Comments
Point 1: There are some recent relevant papers that should be mentioned in the discussion. Most notably the Corso et al. Neoplasia 2018 (PMCID: 5915970) which describes the use of short term rituximab treatment at time of mouse implant which effectively prevents the development of lymphomas in gastric carcinoma PDX implants. Since the authors describe a similar phenomenon, the Corso paper is a critical reference for this manuscript. Other recent published papers that could be referenced include: Ryu et al. Cancers 2019 (PMID: 30965636) and the Liu and Meltzer review paper in Cellular and Molecular Gastroenterology and Hepatology 2017 (PMCID: 5404028).
Response 1: Thank you for providing this information on recent studies. We have referenced these three additional studies in the revised manuscript and also cite the Corso paper in the Discussion section.
Line 359
“More recently, Corso et al. reported the occurrence of lymphoma after gastric cancer tissue engraftment in NOD/SCID mice [29]. We further identified that most of the LPLs were derived from B lymphocytes and were not EBV-associated, although Corso et al. observed that the lymphomas that developed in their study were EBV-associated. We do not deny the potential contribution of EBV to LPLs in the present study, however, as at least one of the cases showed diffuse EBER expression in proliferating B cells.”
Point 2: The authors utilized high percentage fetal calf serum in the preservation media and in the cell culture conditions. It is expected that the FCS inclusion will promote proliferation due to the high growth factor content. The authors should acknowledge or discuss that this could be related to differences between the CDX and PDX development.
Response 2: Thank you for this suggestion. We have added the following information to the Discussion section.
Line 339
“We also note the possibility that the relatively higher concentration of FCS (15%) used in the present study may have promoted the proliferation of more aggressive clones, although further studies are needed to confirm this.”
Point 3: Another discussion point that is underdeveloped here is the paradoxical finding that the well differentiated adenocarcinomas were more readily established than poorly differentiated whereas the highly successful CDX were all poorly differentiated. Moreover, more local-regionally advanced tumors (node positive) were more likely to form PDX. These potentially counter-intuitive findings could be discussed better.
Response 3: Thank you for your suggestion regarding one of the most interesting findings in the present study. The exact reasons for this observation remain to be clarified, but one possibility is that well-differentiated adenocarcinomas are less demanding with respect to stromal interactions. Considering that poorly differentiated adenocarcinoma, especially scirrhous-type carcinoma, utilizes several factors including TGF-beta and FGF for communicating with stromal cells, PDA may have failed to grow in the mouse subcutaneous environment because of a lack of inter-species interaction of some factors. As this is only one possible reason for this finding, with further research needed to clarify the mechanism, we added the following information to the Discussion section.
Line 343
“Although the reason why differentiated adenocarcinoma was more readily established than poorly differentiated adenocarcinoma remains to be elucidated, one possibility is that the former places fewer demands on stromal cells. For example, poorly differentiated adenocarcinoma is known to utilize TGF-beta during stromal reactions, although human TGF-beta may not interact with mouse stromal cells.”
The reason why pN factor but not pM factor was associated with the success rate for PDX establishment remains unknown. One possibility is that the most frequent cause of pM in gastric cancer is peritoneal dissemination, which is more often related to the poorly differentiated adenocarcinoma phenotype. We reanalyzed the data following stratification by histological differentiation grade, but pM again showed no association with PDX success rate, even in the DA subgroup. To clarify this observation, Table 2S and the following information in the Results section were added to the revised manuscript.
Line 300
“Although the reason why pM did not show a significant correlation with the PDX establishment rate remains to be clarified, one possibility is that pM is associated with PDA phenotype as the main cause of pM in gastric cancer is peritoneal dissemination, which is frequently observed in the PDA phenotype. Thus, we also reanalyzed the data after stratification by histological differentiation grade and found that pM again failed to show an association with PDX success rate, even in the DA subgroup (Table S3)”
Table S3 Pathological factors related to PDX establishment (adenocarcinoma cases only, stratified by type of histological differentiation grade)
Histology of primary tumor | pT/N/M | PDX | |||
Fail | Success | p-value | |||
PDA | pT | 1/2 | 15 | 3 | 0.20 |
1 | 66 | 5 | |||
pN | 0 | 23 | 1 | 0.44 | |
1 | 58 | 7 | |||
pM | 0 | 77 | 8 | 1.00 | |
1 | 4 | 0 | |||
DA | pT | 0 | 30 | 6 | 0.10 |
1 | 57 | 12 | |||
pN | 0 | 34 | 2 | 0.03 | |
1 | 53 | 16 | |||
pM | 0 | 86 | 18 | 1.00 | |
1 | 1 | 0 | |||
DA: Differentiated adenocarcinoma, PDA: Poorly differentiated adenocarcinoma
Point 4: It would be good to know how much time occurred between last dose of chemotherapy and surgical resection for those patients who received chemotherapy. There are some PDX groups that recommend a minimum of 3 weeks between chemo administration and tumor cell harvesting for PDX implantation. The authors should provide this info if available.
Response 4: Unfortunately we did not collect data on the duration between the last dose of chemotherapy and surgical resection. In general, as standard procedure in our hospital, chemotherapy was discontinued at least 2 weeks prior to surgical resection.
Point 5: Have the authors considered using a less immunocompromised mouse such as athymic nude mice for the host organism? Perhaps those mice would be less likely to form lymphomas.
Response 5: We used only NOG, NSG and SCID mice as the primary xenograft recipient as we anticipated that a less immunocompromised mouse model may show a reduction in the rate of primary xenograft development.
Point 6: Table 8 needs to be formatted better with words being wrapped into two lines for many of the columns.
Response 6: Thank you for the suggestion; words wrapped into two lines have been abbreviated in the revised manuscript.
Reviewer 3 Report
In this manuscript, the authors summarized their experiences in establishment of patient—derived xenograft (PDX) and cell line generated xenograft (CDX) models and compared pathological morphology of these models from the same patients. Since this is the first publication which describes the pathological differences between primary tumors, PDXs, and CDXs from gastric cancer, the information provided from this manuscript should be useful to the audiences who are also working on PDX and CDX models in their study. There are only a few suggestions to improve the manuscript:
Were the tissue samples obtained before any treatment? Please clarify it. Since the patients’ information were obtained, it would be interesting to know whether success of the engrafting is correlated with any of the tumor characteristics, such as stage, node status, and the treatment in addition to differentiation status.
In Figure 3, it is not clear why the authors used HER2 staining to show differentiation grade. Please briefly explain why HER2 is important in gastric cancer. It is better to show representative cases of DA and PDA here.
In line 233, please spell out EBER and briefly explain why you would like to check EBER.
Author Response
Response to Reviewer 3 Comments
Point 1: Were the tissue samples obtained before any treatment? Please clarify it. Since the patients’ information were obtained, it would be interesting to know whether success of the engrafting is correlated with any of the tumor characteristics, such as stage, node status, and the treatment in addition to differentiation status.
Response 1: Thank you for the suggestions. As we described in Table 7, we evaluated pT, pN and pM factor, and found pN but no other factors were associated with success rate.
Regarding treatment prior to surgical resection, 34 patients received pre-operative chemotherapy. The data was shown in Table 2, and we also explained that “The establishment rate for PDXs was higher in those cases that received chemotherapy than in those that did not (26.4% (9/34) vs. 13.1% (26/198), respectively), although the difference was not statistically significant.” in Results section (line 284).
Point 2: In Figure 3, it is not clear why the authors used HER2 staining to show differentiation grade. Please briefly explain why HER2 is important in gastric cancer. It is better to show representative cases of DA and PDA here.
Response 2: Thank you for the suggestion. Although expression of HER2 is not within the scope of this study, we showed the data because HER2 is important as a molecular target therapy in gastric cancer. Although it will be nice to show HER2 expression in PDA, as you suggested, we have not finished to examine HER2 expressions in CDX.
To explain the importance of HER2 in gastric cancer, we add the following information in Results section
Line 220
“Since HER2 expressions are routinely examined in gastric cancer as the target of Trastuzumab, we investigated HER2 expression in PDX and CDX from the case No.34 which showed HER2 positive in primary tumor. Strong Her2 expressions were observed in PDX and CDX (Figure 3).”
Point 3: In line 233, please spell out EBER and briefly explain why you would like to check EBER..
Response 3: Thank you for the suggestion. We spelled out EBER more accurately in Materials and Methods section, and briefly explained the reason why we examined EBER in .
Line 172
“Epstein-Barr virus (EBV)-encoded small RNAs”
Line 268
“Since EBV is known to transform human B cells, we examined whether the proliferating B cell were EBV positive.”
Reviewer 4 Report
The paper by Kuwata et al. describes the establishment of gastric PDX and cell lines. Other papers have already described the establishment of gastric PDXs, although the number of models was lower.
The paper is clearly written but the information provided is very poor. To verify if the PDX platform described in this paper can be considered representative, more information is needed. For example, the MSI status has not been evaluated and thus it is not clear if among the generated PDXs are included also MSI tumors (which encompass 20% of gastric cancers). This should be added.
Another important point is the lack of information about the anatomic site of origin of the tumor to verify if this is correlated with engraftment. Have tumors of the junction been included?
The pathologic classification used is questionable. Why haven’t the authors used the Lauren’s classification? In previous papers where gastric PDXs have been generated, the difficulty of generating “diffuse type” PDXs has been reported. More pathologic details should be given.
Concerning the Lymphoproliferative disease, the authors say that LPLs were EBER negative. This is not in agreement with what reported by other authors who almost always found a strong positivity for EBV. The authors should try to analyse the EBV status with a different method.
Minor points
Fig. 3 g-i is of very poor quality
Literature citation can be strongly improved. Not so many papers have been published on gastric PDXs and they should be adequately cited.
Author Response
Response to Reviewer 4 Comments
Point 1: The paper is clearly written but the information provided is very poor. To verify if the PDX platform described in this paper can be considered representative, more information is needed. For example, the MSI status has not been evaluated and thus it is not clear if among the generated PDXs are included also MSI tumors (which encompass 20% of gastric cancers). This should be added.
Response 1: Thank you for your comment. How molecular profiling in PDXs/CDXs differs from that in primary tumors is also of interest to us, and we are currently using NGS sequence-based gene mutation analysis to investigate this. Our preliminary findings suggested that the molecular profiles are relatively conserved between primary tumor and PDX. However, the data are based on a small number of cases, and no definitive conclusion has yet been reached. We are also interested in the MSI status of gastric cancer and plan to evaluate this in future studies, but again this is beyond the scope of the current study.
Point 2: Another important point is the lack of information about the anatomic site of origin of the tumor to verify if this is correlated with engraftment. Have tumors of the junction been included?
Response 2: We reanalyzed the data to determine whether tumor location was associated with PDX establishment success rate. Among 198 surgical cases who did not receive prior chemotherapy, 8 cases had esophagogastric-junction cancers. Other cases had tumors located primarily in the upper, middle, and lower portion of the stomach (45, 49, and 93 cases, respectively; tumor location was not available for one case). PDX establishment success rate was highest in esophagogastric-junction cases (20.0%) followed by lower (15.1%), middle (10.2%), and upper (8.9%) locations. These results were considered very interesting, although the mechanism behind these differences remains to be clarified.
This information has been added to the Results section and Table S4 of the revised manuscript.
Line 307
“To investigate the possibility that the location of tumor origin was associated with PDX establishment success, we analyzed the data by tumor location. Interestingly, establishment success rate was highest among esophagogastric-junction cases (20.0%), followed by the lower (15.1%), middle (10.2%), and upper (8.9%) portions of the stomach (Table S4). The mechanism associated with these differences remains to be clarified.”
Table S4 PDX establishment success and tumor location
PDX | |||
Failed | Success | Success rate | |
EGJ | 8 | 2 | 20.0% |
U | 41 | 4 | 8.9% |
M | 44 | 5 | 10.2% |
L | 79 | 14 | 15.1% |
N/A | 0 | 1 | 0.0% |
EGJ: Esophagogastric-junction, U: Upper, M: Middle, L: Lower, N/A: Not available
Point 3: The pathologic classification used is questionable. Why haven’t the authors used the Lauren’s classification? In previous papers where gastric PDXs have been generated, the difficulty of generating “diffuse type” PDXs has been reported. More pathologic details should be given.
Response 3: Although Lauren’s classification is widely accepted in Western countries, we used the Japanese classification of gastric carcinoma criteria, which is consistent with WHO classification and is generally used for pathological diagnosis in Japan. As the clinicopathological data were extracted from the electronic medical record system in our hospital, these criteria had been implemented. In addition, in the present study, we examined morphology in relatively small sized PDX and CDX tumors, and this preferred to use histological subtyping based on differential grade.
Information on classification has been added to the Materials and Methods section, including the citation of the Japanese classification of gastric carcinoma criteria.
Line 165
Histology, Immunohistochemistry (IHC), and in situ hybridization (ISH)
Tissue sections (4- mm slices) were prepared and subjected to H.E. staining, IHC, and ISH. Histological classifications were made based on the Japanese classification of gastric carcinoma (3rd English edition) and differentiation grade was based on the presence/absence of glandular structure formation.
Sano, T.; Aiko, T. New Japanese classifications and treatment guidelines for gastric cancer: revision concepts and major the revised points. Gastric Cancer 2011, 14, 97-100.
Point 4: Concerning the Lymphoproliferative disease, the authors say that LPLs were EBER negative. This is not in agreement with what reported by other authors who almost always found a strong positivity for EBV. The authors should try to analyse the EBV status with a different method.
Response 4: We acknowledge previous reports indicating the presence of EBV in LPL development in gastric cancer xenograft models. However, we observed that EBV-positive LPLs were rare in our study, and cannot explain the lack of agreement with previous studies. Since the detection threshold may have caused this inconsistency, it may be beneficial to utilize a different method. However, the cause of LPLs was not the focus of the present study, so it was deemed not appropriate to evaluate an alternative method during manuscript revision.
As we do not deny the possibility that EBV might have contributed to the LPLs to some extent, particularly because one case in the present study showed diffuse EBER positivity in proliferating B cells, we have added the following sentence to the Discussion section.
Line 363
“We do not deny the potential contribution of EBV in to LPLs in the present study, however, as at least one of the cases showed diffuse EBER expression in proliferating B cells.”
Point 5: Fig. 3 g-i is of very poor quality
Response 5: We have re-captured the HER2-DISH slide images at higher magnification and replaced the images in the original manuscript in Figure 3 g-i (Line 224 )
Point 6: Literature citation can be strongly improved. Not so many papers have been published on gastric PDXs and they should be adequately cited.
Response 6: We have added the following three recent references (including one review paper) to the reference list:
Corso, S.; Cargnelutti, M.; Durando, S.; Menegon, S.; Apicella, M.; Migliore, C.; Capeloa, T.; Ughetto, S.; Isella, C.; Medico, E., et al. Rituximab Treatment Prevents Lymphoma Onset in Gastric Cancer Patient-Derived Xenografts. Neoplasia 2018, 20, 443-455.
Ryu, W.J.; Lee, J.E.; Cho, Y.H.; Lee, G.; Seo, M.K.; Lee, S.K.; Hwang, J.H.; Min, D.S.; Noh, S.H.; Paik, S., et al. A Therapeutic Strategy for Chemotherapy-Resistant Gastric Cancer via Destabilization of Both beta-Catenin and RAS. Cancers (Basel) 2019, 11, 496.
Liu, X.; Meltzer, S.J. Gastric Cancer in the Era of Precision Medicine. Cell Mol Gastroenterol Hepatol 2017, 3, 348-358.
Reviewer 5 Report
In this study Kuwata et al describe the establishment and characterization of gastric cancer patient derived xenografts. I think this is a well executed and well written study. I have included some questions and suggestions for improvement.
Can the authors elaborate on the methods used to isolate tumour cells from fibroblasts in culture OR provide a reference.
Can the authors comment on and provide data detailing the time it took for tumours to establish (to palpable stage) at each passage
Page 6 Line 204 Table 3 “It is noteworthy that more than half the cases in which primary tumors showed differentiated histology (differentiated adenocarcinomas, DAs) became poorly differentiated adenocarcinomas (PDAs) in CDXs.” Conversely in the PDX approx. 50% of the PDA primary tumours became DA when implanted as a PDX. Can the authors comment on this?
Did any of the patients have evidence of metastatic disease (either before or subsequent to surgery)? If so do the mice implanted with PDX or cell lines also develop metastases?
I am happy to see that the authors checked for B-cell transformation. This is a phenomenon we have encountered in our laboratory and is rarely reported!
The authors suggest that PDX model systems will be useful to test new and emerging drug treatments. Given the advances in immunotherapies could the authors include a section on the limitations of using these types of model systems in such a setting?
Author Response
Response to Reviewer 5 Comments
Point 1: Can the authors elaborate on the methods used to isolate tumour cells from fibroblasts in culture OR provide a reference.
Response 1: We have amended the applicable text in the Materials and Methods section as follows.
Line 144
“For establishing the cell lines from primary or xenograft tumors, tumor tissue was dissected into 1-mm cubic squares and explanted into 60-mm Corning Primaria dishes (Corning, NY, USA). An additional 4 ml of culture medium was added the next day, and 50% of the culture medium was replaced twice weekly. Dishes containing tissue fragments were observed daily under an inverted phase microscope. The following three methods were used to selectively remove overgrowth fibroblasts: (i) trypsin treatment (0.05% trypsin and 0.02% EDTA, Thermo Fisher Scientifics): fibroblasts exfoliation by differences in trypsin sensitivity, with fresh medium added and washing performed to remove fibroblasts; (ii) physical treatment: change to serum-free medium and detachment of fibroblasts only using sharp silicone rubber under a microscope; and (iii) after exfoliating the cells using enzymatic treatment (Tumor Dissociation Kit # 130-095-929, Miltenyi Biotec, Tokyo, Japan), mouse-derived cells were removed using an antibody column (Mouse Cell Depletion Kit # 130-104-694, Miltenyi Biotec) according to the manufacturer’s protocol.”
Point 2: Can the authors comment on and provide data detailing the time it took for tumours to establish (to palpable stage) at each passage
Response 2: We reanalyzed the data and calculated the time it took for tumors to establish at each passage. The relevant information has been added to the Results section, and is shown in Table S1.
Line 186
“The average number of weeks of tumor growth for xenograft expansion until tumor harvesting is shown in Table S1.”
Table S1 Average number of weeks of tumor growth for xenograft expansion until tumor harvesting
Histology of primary tumor | n | Generation of xenograft: mean ± SD (min–max) | ||||
1st | 2nd | 3rd | 4th | 5th | ||
DA | 27 | 17.4 ± 14.8 | 10.4 ± 7.7 | 7.4 ± 3.8 | 8.8 ± 7.0 | 7.2 ± 3.8 |
(4.3–58.1) | (3.1–35.1) | (2.9–16.0) | (2.1–35.0) | (2.4–19.7) | ||
PDA | 8 | 16.8 ± 15.6 | 20.2 ± 18.0 | 14.6 ± 11.5 | 13.2 ± 8.8 | 9.5 ± 5.5 |
(8.4–57.9) | (4.0–59.7) | (5.0–42.0) | (5.9–33.0) | (4.1–21.0) | ||
DA: Differentiated adenocarcinoma, PDA: Poorly differentiated adenocarcinoma
Point 3: Page 6 Line 204 Table 3 “It is noteworthy that more than half the cases in which primary tumors showed differentiated histology (differentiated adenocarcinomas, DAs) became poorly differentiated adenocarcinomas (PDAs) in CDXs.” Conversely in the PDX approx. 50% of the PDA primary tumours became DA when implanted as a PDX. Can the authors comment on this?
Response 3: Thank you for raising this interesting point. As described in the manuscript, most cases showing discordant histology between primary tumors and PDX were of mixed differential grade. It should also be noted that tissue samples were primarily obtained from the mucosal/superficial region, whereas differentiated histology was most frequently observed in adenocarcinoma with mixed histology.
We have added the following information to the Results section.
Line 217
“It should also be noted that tissue samples for PDX were primarily obtained from the mucosal/superficial areas of tumor lesions. In adenocarcinoma with mixed histology, differentiated histology is most frequently observed in the superficial area.”
Point 4: Did any of the patients have evidence of metastatic disease (either before or subsequent to surgery)? If so do the mice implanted with PDX or cell lines also develop metastases?
Response 4: Some of the patients in the surgery group had metastatic lesions before and /or after surgical resection. All of the patients in the CART group had peritoneal dissemination. However, no evidence of PDX/CDX metastatic lesions have been observed to date in the mouse models.
To emphasize this finding, we have added the following text to the Discussion section.
Line 372
“Third, none of the PDX or CDX models developed metastatic lesions in mice. Therefore, the models are not suitable for examining the mechanism of tumor metastasis.”
Point 5: I am happy to see that the authors checked for B-cell transformation. This is a phenomenon we have encountered in our laboratory and is rarely reported!
Response 5: Thank you very much for this encouraging comment.
Point 6: The authors suggest that PDX model systems will be useful to test new and emerging drug treatments. Given the advances in immunotherapies could the authors include a section on the limitations of using these types of model systems in such a setting?
Response 6: Thank you for this suggestion. How we can apply our PDX/CDX models for immunotherapy remains one of our most important questions and challenges. We have added the following sentence as a limitation in the Discussion.
Line 374
“Fourth, since immunodeficient mice are required for the development of PDX/CDX models, these models are not suitable for the evaluation of immunotherapies unless a humanized immune system is restored by co-transplantation of immune cells from the same patient.”
Round 2
Reviewer 4 Report
As the authors claim that they have generated a PDX platform that is representative of gastric gancer, MSI evaluation is mandatory.